# Be Tangential to Manifold: Discovering Riemannian Metric for Diffusion Models

## Abstract

Diffusion models are powerful deep generative models (DGMs) that generate high-fidelity, diverse content. However, unlike classical DGMs, they lack an explicit, tractable low-dimensional latent space that parameterizes the data manifold. This absence limits manifold-aware analysis and operations, such as interpolation and editing. Existing interpolation methods for diffusion models typically follow paths through high-density regions, which are not necessarily aligned with the data manifold and can yield perceptually unnatural transitions. To exploit the data manifold learned by diffusion models, we propose a novel Riemannian metric on the noise space, inspired by recent findings that the Jacobian of the score function captures the tangent spaces to the local data manifold. This metric encourages geodesics in the noise space to stay within or run parallel to the learned data manifold. Experiments on image interpolation show that our metric produces perceptually more natural and faithful transitions than existing density-based and naive baselines.

## 1 Introduction

Diffusion models are a class of deep generative models (DGMs) that have shown a remarkable capability to generate high-fidelity, diverse content (Ho et al., 2020; Song et al., 2021a; Rombach et al., 2022). They can be applied to various downstream tasks, including interpolation, inversion, and editing (Hertz et al., 2023; Mokady et al., 2023; Danier et al., 2024). Theoretical investigation can help the understanding of their mechanisms and enhance their applicability.

The *manifold hypothesis* has long played a central role in the theoretical analysis of DGMs, such as variational autoencoders (VAEs) (Kingma & Welling, 2014) and generative adversarial networks (GANs) (Goodfellow et al., 2014). This hypothesis states that real-world data (e.g., images) are concentrated around a low-dimensional manifold embedded in the high-dimensional data space (Bengio et al., 2012; Fefferman et al., 2016). In this context, DGMs are understood to learn not only the data distribution but also its underlying manifold, either explicitly or implicitly (Loaiza-Ganem et al., 2024). In VAEs and GANs, the latent space is interpreted as a parameterization of this data manifold (Arjovsky & Bottou, 2017). Various studies leverage this geometric perspective to analyze the learned structure and improve generation quality (Gruffaz & Sassen, 2025). One example is to introduce a Riemannian metric on the latent space by pulling back the metric on the data space through the decoder. This enables geometrically meaningful operations within the latent space. For example, traversing the latent space along geodesics yields interpolations that are faithful to the intrinsic geometric structure of the data (Shao et al., 2017; Arvanitidis et al., 2018; Chen et al., 2018; Arvanitidis et al., 2021).

Unlike VAEs or GANs, diffusion models lack an explicit low-dimensional latent space, which complicates the direct application of conventional pullback-based geometric approaches. Interpolations are typically realized as paths that traverse linearly or through high-density regions of the learned data distribution at an intermediate generation step (i.e., in the noise space) (Samuel et al., 2023; Zheng et al., 2024; Yu et al., 2025). We provide a conceptual illustration in Fig. 1. These approaches, however, are not necessarily aligned with the intrinsic geometry of the data manifold and often lead to visually unnatural and abrupt transitions (e.g., over-smoothed). This is because a linear path may cut through low-density regions, and a high-density path may lose the characteristics of endpoints (Karczewski et al., 2025a).

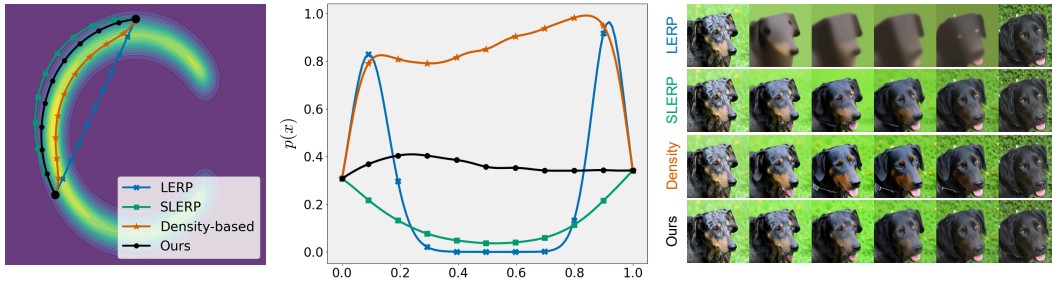

Figure 1: **A conceptual comparison of interpolation.** (left) Interpolation paths on a C-shaped distribution. (middle) A plot of the probability density transitions for their corresponding interpolation paths. (right) Examples of image interpolation on Animal Faces-HQ (AF) (Choi et al., 2020). LERP cuts through a low-density region, yielding unnatural transitions. SLERP deviates from the manifold, sometimes losing detail textures (see the background in the right panel). Density-based interpolation approaches and traverses a high-density region, not preserving the probabilities of the endpoints and sometimes producing over-smoothed images. **Ours** runs parallel to the manifold, preserving the probabilities of the endpoints and yielding natural transitions. See Section 5 for details.

To characterize the data geometry learned by diffusion models, we propose a novel Riemannian metric on the noise space derived from the Jacobian of the score function. The contribution of this work is threefold. **(i) Riemannian metric for the noise space.** With our metric, we can treat the noise space of a pre-trained diffusion model as a Riemannian manifold without any further training or architectural modifications. **(ii) Manifold-aware geodesics.** The construction of our metric is motivated by recent findings that degeneracy in the Jacobian of the score function captures the local structure of the data manifold (Stanczuk et al., 2024; Ventura et al., 2025). Our metric encourages geodesics to stay within or run parallel to the data manifold. **(iii) Empirical interpolation performance.** We validate our approach on synthetic data interpolation, image interpolation, and video frame interpolation. Results demonstrate that our metric yields perceptually more natural and faithful transitions than existing density-based methods and naive baselines.

## 2 RELATED WORK

**Latent Space Manipulation in Deep Generative Models.** The manifold hypothesis states that real-world data (e.g., images) lie on a low-dimensional manifold embedded in a high-dimensional data space (Bengio et al., 2012; Fefferman et al., 2016), where DGMs are understood to learn this data manifold (Loaiza-Ganem et al., 2024). In VAEs and GANs, the latent space parameterizes the data manifold, and the decoder (or generator) embeds this manifold in data space as the image of the latent space (Arjovsky & Bottou, 2017). This structure implies that manipulating latent variables traverses the data manifold and ensures generated outputs to remain semantically coherent (Ramesh et al., 2019). Indeed, linear traversals in latent space have become a common approach for editing the semantic attributes of generated images (Goetschalckx et al., 2019; Härkönen et al., 2020; Plumerault et al., 2020; Shen et al., 2020; Voynov & Babenko, 2020; Oldfield et al., 2021; Shen & Zhou, 2021; Spingarn et al., 2021; Zhuang et al., 2021; Haas et al., 2022). However, as real-world data distributions are skewed and heterogeneous, linear manipulations often encounter limitations in quality. While non-linear approaches improve editing quality, they often require training additional networks and can distort the learned manifold structure (Ramesh et al., 2019; Jahanian et al., 2020; Tewari et al., 2020; Abdal et al., 2021; Khrulkov et al., 2021; Liang et al., 2021; Tzelepis et al., 2021; Chen et al., 2022; Choi et al., 2022; Aoshima & Matsubara, 2023).

**Riemannian Geometry of Deep Generative Models.** Applying ideas from Riemannian geometry to the latent spaces of DGMs is an active area of research (Gruffaz & Sassen, 2025). Some methods require training additional networks (Yang et al., 2018; Arvanitidis et al., 2022; Lee et al., 2022; Sorrenson et al., 2025). Another common approach is to construct the pullback metric by pulling back the Euclidean metric from the data space through the decoder of a pre-trained model (Shao

et al., 2017; Chen et al., 2018; Arvanitidis et al., 2018; 2021). This enables leveraging the geometric structure learned by the model without additional training.

**Interpolation in Diffusion Models.** Diffusion models learn a denoising function, which iteratively denoises noisy samples backward in time from $t = T$ to $t = 0$ and obtains clean sample at $t = 0$, thereby forming the data distribution (Sohl-Dickstein et al., 2015; Ho et al., 2020; Song et al., 2021a;b; Rombach et al., 2022). A space of noisy samples at $t > 0$ is often referred to as a *noise space*. Unlike VAEs or GANs, diffusion models lack an explicit low-dimensional latent space, yet empirical observations show that the noise space acts as a latent space (Ho et al., 2020). However, the iterative nature of the generation process makes it difficult to define a pullback metric.

Earlier works employ linear interpolation (LERP), which interpolates noisy samples linearly in noise space (Ho et al., 2020). However, LERP often degrades perceptual quality in interpolated images, as shown in Fig. 1. Noisy samples at time $t = T$ are typically drawn from a standard Gaussian prior and therefore concentrate on a hypersphere with radius approximately $\sqrt{D}$, where $D$ denotes the dimensionality. LERP between two noisy samples produces interpolated points with unnaturally small vector norms, losing detailed features. A similar trend holds for interpolations at intermediate timesteps $t < T$. Spherical linear interpolation (SLERP) addresses this issue by interpolating noisy samples along the surface of a hypersphere, preserving the norms of noisy samples (Shoemake, 1985; Song et al., 2021a). Other approaches also leverage the norm density of the Gaussian prior at $t = T$ (Samuel et al., 2023) or attempt to preserve the variance of pixel intensity (Bodin et al., 2025). However, empirically, noised real samples do not follow a Gaussian distribution even at $t = T$, degrading the interpolation quality in practice (Zheng et al., 2024).

Some studies treat an intermediate layer of the neural networks used in diffusion models as a latent space, such as the bottleneck layer (Kwon et al., 2023; Park et al., 2023a;b) of U-Nets (Ronneberger et al., 2015) and the attention layer (He et al., 2024) of Vision Transformers (Dosovitskiy et al., 2020). However, these neural networks employ skip connections that allow information to bypass other layers, which hinders the models from generating new samples only from these surrogate latent spaces. Various studies have explored specialized architectures and additional training for image interpolation (Preechakul et al., 2022; Zhang et al., 2023; Wang & Golland, 2023; Guo et al., 2024; Lu et al., 2024; Shen et al., 2024; Yang et al., 2024; Kim et al., 2025; Lobashev et al., 2025), whereas we focus on investigating the geometric structure learned by a diffusion model itself without any further training or architectural modifications.

**Density-based Interpolation in Diffusion Models.** Other methods leverage the noisy-sample density at intermediate timesteps $t < T$. GeodesicDiffusion (Yu et al., 2025) defines a conformal metric by multiplying by the inverse density of noisy samples, guiding interpolated images to lie in high-density regions. This approach is also common in other DGMs (Rezende & Mohamed, 2015; Du & Mordatch, 2019), such as normalizing flows (Sorrenson et al., 2025) and energy-based models (Béthune et al., 2025). Other studies have also proposed to prioritize high-density regions by designing metrics (Azeglio & Bernardo, 2025). However, recent studies have shown that image likelihood is negatively correlated with perceptual detail: images in high-density regions are often over-smoothed and lose detailed features, whereas images in lower-density regions may contain richer textures and fine-grained details (Karczewski et al., 2025a). This observation shows the limitations of interpolations based on high-density paths. Although some studies draw inspiration from statistical manifolds, it remains unclear what structures their methods leverage (Karczewski et al., 2025b; Lobashev et al., 2025).

**Data Manifold in Diffusion Models.** Diffusion models have been shown to implicitly learn the data manifold (Pidstrigach, 2022; Wenliang & Moran, 2022; Tang & Yang, 2024; George et al., 2025; Potaptchik et al., 2025). Methods based on high-density regions assume that such regions correspond to the data manifold. From a different perspective, several studies have attempted to estimate the local intrinsic dimension of the data manifold (Horvat & Pfister, 2024; Kamkari et al., 2024; Stanczuk et al., 2024; Humayun et al., 2025; Ventura et al., 2025). Their key insight is that the rank deficiency of the Jacobian of the score function (i.e., the Hessian of the log-density) equals the dimension of the data manifold (Stanczuk et al., 2024; Ventura et al., 2025). We build upon this insight to define a Riemannian metric on the noise space of a pre-trained diffusion model.

## 3 PRELIMINARIES

### 3.1 RIEMANNIAN GEOMETRY

**Riemannian metric.** We adopt the notions in Lee (2019). Let $\mathcal{M}$ be a smooth manifold. A *Riemannian metric* $g$ on $\mathcal{M}$ is a smooth covariant 2-tensor field such that, at every point $p \in \mathcal{M}$, the tensor $g_p$ defines an inner product on the tangent space $T_p\mathcal{M}$. A *Riemannian manifold* is the pair $(\mathcal{M}, g)$. Using local coordinates, the metric $g_p$ can be expressed as a symmetric and positive definite matrix $G_p$ at $p$. See Appendix A.1 for this connection. The inner product $\langle v, w \rangle_g$ of two tangent vectors $v, w \in T_p\mathcal{M}$ at $p$ is given by

$$\langle v, w \rangle_g = g_p(v, w) = v^\top G_p w.$$

**Geodesics.** The length of a tangent vector $v \in T_p\mathcal{M}$ is given by $|v|_g := \sqrt{\langle v, v \rangle_g}$. For a smooth curve $\gamma : [0, 1] \to \mathcal{M}$, $u \mapsto \gamma(u)$, its length is

$$L[\gamma] := \int_0^1 |\gamma'(u)|_g \mathrm{d}u = \int_0^1 \sqrt{\langle \gamma'(u), \gamma'(u) \rangle_g} \mathrm{d}u = \int_0^1 \sqrt{\gamma'(u)^\top G_{\gamma(u)} \gamma'(u)} \mathrm{d}u. \tag{1}$$

A *geodesic* is a curve that locally minimizes length; intuitively, it is a locally shortest path between two points. It is often more convenient to work with the energy functional $E[\gamma]$:

$$E[\gamma] = \tfrac{1}{2} \int_0^1 |\gamma'(u)|_g^2 \, \mathrm{d}u = \tfrac{1}{2} \int_0^1 \langle \gamma'(u), \gamma'(u) \rangle_g \, \mathrm{d}u. \tag{2}$$

Any constant-speed geodesic is a critical point of the energy functional.

### 3.2 DIFFUSION MODELS

**Forward Process.** Let $x_0 \in \mathbb{R}^D$ be a data sample. The forward process is defined as a Markov chain which adds Gaussian noise at each timestep $t = 1, \ldots, T$ recursively:

$$q(x_t|x_{t-1}) = \mathcal{N}\left(x_t; \sqrt{1 - \beta_t}x_{t-1}, \beta_t I\right) = \mathcal{N}\left(\sqrt{\tfrac{\alpha_t}{\alpha_{t-1}}}x_{t-1}, \left(1 - \tfrac{\alpha_t}{\alpha_{t-1}}\right)I\right), \tag{3}$$

where $\{\beta_t\}_{t=1}^T$ is a scheduled variance, $I$ is the identity matrix in $\mathbb{R}^D$, and $\alpha_t = \prod_{s=1}^t (1 - \beta_s)$. $x_t$ becomes progressively more corrupted by noise as $t$ increases, and $x_T$ is nearly an isotropic Gaussian distribution.

**Reverse Process.** The generation process of diffusion models is referred to as the reverse process, which inverts the forward process by iteratively denoising a noisy sample $x_T \sim \mathcal{N}(0, I)$ backward in time from $t = T$ to $t = 0$ and obtaining a clean sample $x_0$. Namely, a reverse Markov chain $p_t(x_{t-1}|x_t; \theta)$ is constructed as

$$x_{t-1} = \tfrac{1}{\sqrt{1 - \beta_t}}\left(x_t - \tfrac{\beta_t}{\sqrt{1 - \alpha_t}}\epsilon_\theta(x_t, t)\right) + \sigma_t z_t, \tag{4}$$

with a trainable noise predictor $\epsilon_\theta$, where $z_t \sim \mathcal{N}(0, I)$, and $\sigma_t^2 = \beta_t$ is a variance at timestep $t$. The noise predictor $\epsilon_\theta(x_t, t)$ is trained by minimizing the objective:

$$\mathcal{L}(\theta) = \mathbb{E}_{x, \epsilon_t, t}\left[\|\epsilon_t - \epsilon_\theta(x_t, t)\|_2^2\right], \tag{5}$$

where $\epsilon_t \sim \mathcal{N}(0, I)$ is the noise added during the forward process at timestep $t$.

**Denoising Diffusion Implicit Models and Inversion.** Denoising diffusion implicit models (DDIMs) (Song et al., 2021a) modifies Eq. (3) to be a non-Markovian process $q(x_{t-1}|x_t, x_0) = \mathcal{N}(\sqrt{\alpha_{t-1}}x_0 + \sqrt{1 - \alpha_{t-1} - \sigma_t^2}, \sigma_t^2 I)$. Then, the reverse process becomes

$$x_{t-1} = \sqrt{\alpha_{t-1}}\left(\tfrac{x_t - \sqrt{1 - \alpha_t}\epsilon_\theta(x_t, t)}{\sqrt{\alpha_t}}\right) + \sqrt{1 - \alpha_{t-1} - \sigma_t^2} \cdot \epsilon_\theta(x_t, t) + \sigma_t z_t, \tag{6}$$

where $\sigma_t = \eta\sqrt{(1 - \alpha_{t-1})/(1 - \alpha_t)}\sqrt{1 - \alpha_t/\alpha_{t-1}}$. $\eta \in [0, 1]$ controls the stochasticity: $\eta = 1$ recovers DDPM, while $\eta = 0$ yields a deterministic update. The forward process in Eq. (3) can also be modified accordingly. Then, we can deterministically map a clean sample $x_0$ to a noisy sample $x_t$, operate interpolations in the noise space at timestep $t$, and then map it back to a clean sample $x_0$; this procedure is often referred to as DDIM Inversion. See Appendix B.1 for details.

**Formulation as Stochastic Differential Equations.** As the timestep size approaches zero, the forward process can also be formulated as a stochastic differential equation (SDE) (Song et al., 2021b). The reverse process is the corresponding reverse-time SDE that depends on the score function $s_\theta(x_t, t) := \nabla_{x_t} \log p_t(x_t; \theta)$, where $p_t(x_t; \theta)$ denotes the density of $x_t$ at time $t$. Notably, the noise predictor $\epsilon_\theta$ is closely tied to the score function (Luo, 2022) as:

$$s_\theta(x_t, t) = \nabla_{x_t} \log p_t(x_t; \theta) \approx -\epsilon_\theta(x_t, t)/\sqrt{1 - \alpha_t}. \tag{7}$$

Thus, learning the noise predictor $\epsilon_\theta$ is essentially learning the score function $s_\theta$. The following discussion about the score function $s_\theta$ applies to the noise predictor $\epsilon_\theta$ as well, up to a known scale.

**Conditioning and Guidance** We can condition the score function $s_\theta$ on a text prompt $c$, writing $s_\theta(x_t, t, c)$, to guide the generation process (Rombach et al., 2022). The actual implementation depends on the architecture of the score function $s_\theta$. Classifier-Free Guidance (CFG) amplifies this guidance (Ho & Salimans, 2021), and a negative prompt $c_{\text{neg}}$ suppresses certain concepts (Rombach et al., 2022). With these methods, the score function is replaced with

$$\tilde{s}_\theta(x_t, t, c, c_{\text{neg}}) = (w + 1)s_\theta(x_t, t, c) - w s_\theta(x_t, t, \varnothing) - w_{\text{neg}} s_\theta(x_t, t, c_{\text{neg}}), \tag{8}$$

where $s_\theta(x_t, t, c)$, $s_\theta(x_t, t, \varnothing)$, and $s_\theta(x_t, t, c_{\text{neg}})$ are score functions conditioned on the prompt $c$, on no prompts $\varnothing$, and on the negative prompt $c_{\text{neg}}$, respectively. $w \geq 0$ is the guidance scale that amplifies the effect of the condition $c$, and $w_{\text{neg}} \geq 0$ is the scale for the negative prompt $c_{\text{neg}}$.

## 4 METHOD

**Proposed Metric.** Let $x_t$ be a point in the noise space $\mathbb{R}^D$ at time $t$, and $v, w \in T_{x_t}\mathbb{R}^D$ be tangent vectors at $x_t$. We propose a Riemannian metric on the noise space of diffusion models at time $t$ as

$$g_{x_t}(v, w) := \langle J_{x_t} v, J_{x_t} w \rangle = v^\top G_{x_t} w, \tag{9}$$

where $J_{x_t} = \nabla_{x_t} s_\theta(x_t, t)$ is the Jacobian of the score function $s_\theta(\cdot, t)$ (or equivalently, of the noise predictor $\epsilon_\theta(\cdot, t)$ up to scaling), and $G_{x_t} = J_{x_t}^\top J_{x_t}$ is the matrix notion of the metric $g_{x_t}$ at $x_t$. This construction ensures that $G_{x_t}$ is symmetric and positive semidefinite, making it a (possibly degenerate) Riemannian metric. Indeed, the Jacobian $J_{x_t}$ is degenerate on clean data lying a low-dimensional manifold. Moreover, the score function $s_\theta$ is typically not well trained outside the data manifold at time $t = 0$, making it difficult to define a meaningful metric off-manifold (Yu et al., 2025). Hence, we use this metric in the noise space for $t > 0$, where samples are corrupted by noise, the Jacobian $J_{x_t}$ is typically full-rank, and the metric $g_{x_t}$ is positive definite.

To ensure positive definiteness, we can also consider a regularized metric $G_{x_t} = J_{x_t}^\top J_{x_t} + \lambda I$ for a small $\lambda > 0$. However, preliminary experiments using Stable Diffusion v2.1-base (Rombach et al., 2022) showed that this modification does not significantly affect the results, so we use the simpler form in Eq. (9) in the following.

**Interpretation.** Stanczuk et al. (2024) found that as time $t$ approaches zero, the score function $s_\theta(x_t, t)$ points orthogonally towards the data manifold containing the data point $x_t$. Hence, the inner product between the score function $s_\theta(x_t, t)$ and a vector $v$ at $x_t$ is small if $v$ lies in the tangent space to the data manifold, while it is large if $v$ points in the normal direction. Ventura et al. (2025) investigated the Jacobian $J_{x_t}$ of the score function $s_\theta(x_t, t)$ and observed that its rank deficiency corresponds to the dimension of the data manifold when samples are clean and lie on a low-dimensional manifold; for real-world noisy samples, $J_{x_t}$ is typically full-rank but exhibits a sharp spectral gap. Intuitively, the Jacobian $J_{x_t}$ shrinks along tangent directions and remains large along normal directions. More precisely, let $\mathcal{M}_t$ be the data manifold at time $t$ learned by a diffusion model, and $x \in \mathcal{M}_t$ be a point on the manifold $\mathcal{M}_t$. Define the tangent space $\mathcal{T}_x\mathcal{M}_t$ to the manifold $\mathcal{M}_t$ as the $d$-dimensional subspace ($d \ll D$) spanned by the right singular vectors of $J_x$ corresponding to small singular values; the normal space $\mathcal{N}_x\mathcal{M}_t$ is the orthogonal complement spanned by the right singular vectors corresponding to large singular values. Then, the tangent space $T_x\mathbb{R}^D$ to the noise space $\mathbb{R}^D$ at $x$ is decomposed as $T_x\mathbb{R}^D = \mathcal{T}_x\mathcal{M}_t \oplus \mathcal{N}_x\mathcal{M}_t$, and we have:

**Proposition 1.** *Minimizing $\|J_{x_t}v\|_2^2$ with respect to a vector $v$ of a fixed Euclidean norm encourages the vector $v$ to lie in the tangent space $\mathcal{T}_x\mathcal{M}_t$.*

See Appendix A.2 for detailed explanation. Therefore, our proposed metric yields geodesics to follow the tangent directions and stay within the data manifold. When a sample $x_t$ does not lie exactly on the data manifold $\mathcal{M}_t$, the metric still encourages geodesics to run parallel to the data manifold $\mathcal{M}_t$. By contrast, density-based metrics encourage geodesics to approach high-density regions, which may lose fine-grained details and lead to unnatural transitions, as illustrated in Fig. 1.

From another viewpoint, our proposed metric can be interpreted as the pullback $s_\theta^* I$ of the Euclidean metric $I$ on the score space $\mathbb{R}^D$ through the score function $s_\theta$, since $v^\top G_{x_t} w = (J_{x_t} v)^\top I J_{x_t} w$ and $J_{x_t} v, J_{x_t} w \in T_{s_\theta(x_t,t)} \mathbb{R}^D$. A curve $\gamma$ is a geodesic in the noise space under our proposed metric if the score function $s_\theta$ is locally an immersion and maps the curve $\gamma$ to a straight line $s_\theta(\cdot, t) \circ \gamma$. Geodesics under our proposed metric minimize the change in the score function $s_\theta$ along the paths. Earlier studies have shown that gradients of log-likelihoods (with respect to model parameters) can serve as robust, semantically meaningful representations of inputs (Yeh et al., 2018; Charpiat et al., 2019; Hanawa et al., 2021). In this light, our proposed metric can be viewed as a measure of the semantic closeness captured by the score function $s_\theta$ between infinitesimally different samples, providing transitions that preserve the underlying semantics.

**Geodesics for Interpolation.** An interpolation between two points $x_t^{(0)}$ and $x_t^{(1)}$ is considered to be realized as a geodesic path between them. A geodesic can be obtained by solving a second-order ordinary differential equation (ODE) called the geodesic equation (Lee, 2019), which requires computation of $O(D^3)$ in general, not feasible in high-dimensional spaces. Hence, we employ numerical methods to find a geodesic path between two points as a critical point of the energy functional in Eq. (2).

Let $u \in [0, 1]$ be the independent variable that parameterizes a curve $\gamma : u \mapsto \gamma(u)$. The energy functional $E[\gamma]$ in Eq. (2) with our proposed metric in Eq. (9) becomes:

$$E[\gamma] = \frac{1}{2} \int_0^1 \langle J_{\gamma(u)} \gamma'(u), J_{\gamma(u)} \gamma'(u) \rangle \mathrm{d}u = \frac{1}{2} \int_0^1 \| J_{\gamma(u)} \gamma'(u) \|_2^2 \mathrm{d}u = \frac{1}{2} \int_0^1 \| \frac{\partial}{\partial u} s_\theta(\gamma(u), t) \|_2^2 \mathrm{d}u,$$
(10)

where the last equality follows from the chain rule. We discretize the curve $\gamma$ as a sequence of $N+1$ points $x_t^{(0)}, \ldots, x_t^{(1)}$, where $u_0 = 0$, $u_N = 1$, $\Delta u = u_{i+1} - u_i = 1/N$ for $i = 0, \ldots, N-1$, and $x_t^{(u_i)} = \gamma(u_i)$ for $i = 0, \ldots, N$. Then, the energy functional in Eq. (10) is approximated as:

$$E[\gamma] \approx \frac{1}{2} \sum_{i=0}^{N-1} \| (s_\theta(x_t^{(u_{i+1})}, t) - s_\theta(x_t^{(u_i)}, t)) \|_2^2 / \Delta u.$$
(11)

Given two samples $x_t^{(0)}$ and $x_t^{(1)}$, the geodesic path between them is obtained by minimizing the discrete approximation to $E[\gamma]$ in Eq. (11) with respect to the intermediate points $x_t^{(u_1)}, \ldots, x_t^{(u_{N-1})}$. Then, $x_t^{(u_1)}, .., x_t^{(u_{N-1})}$ serve as interpolated samples.

In practice, given a pair of clean samples $x_0^{(0)}$ and $x_0^{(1)}$, we first map them to noisy samples $x_t^{(0)}$ and $x_t^{(1)}$ using DDIM Inversion, then compute the geodesic path between them in the noise space at time $t$ by minimizing Eq. (11), and finally map the interpolated noisy samples $x_t^{(u)}$ back to clean samples $x_0^{(u)}$ using the deterministic reverse process in Eq. (6).

**Limitations and Generalization.** At $t = 0$, the minimization of Eq. (11) may fail to converge properly. This is because the score function $s_\theta$ is not well trained outside the data manifold $\mathcal{M}_0$, and even when it is well trained, if the data manifold $\mathcal{M}_0$ is truly low dimensional, the Jacobian $J_{x_t}$ can be degenerate, and the metric $g_{x_t}$ becomes degenerate as well. Most importantly, since a geodesic is only a local minimizer, a reasonably good initialization of the path is required. For these reasons, we primarily use our proposed metric $g_{x_t}$ in the noise space for $t > 0$. In this setting, samples $x_t$ are corrupted by noise; the Jacobian $J_{x_t}$ is typically full-rank, and our proposed metric $g_{x_t}$ is positive definite. See also Appendix A for details.

Diffusion models learn the score function $s_\theta$ directly rather than the log-density $\log p_t$. Consequently, its Jacobian $J_{x_t}$ need not be symmetric, and a clean decomposition into tangent and normal subspaces $\mathcal{T}_{x_t} \mathcal{M}_t \oplus \mathcal{N}_{x_t} \mathcal{M}_t$ is not guaranteed at a point $x_t \in \mathcal{M}_t$. Even then, the Jacobian $J_{x_t}$ typically exhibits a sharp spectral gap, and Proposition 1 still holds approximately. Since diffusion models are often used with CFG or negative prompts, we replace the score function $s_\theta$ with the guided update in Eq. (8) when needed. The resulting metric then reflects the manifold of data generated by the guided model. In all cases, the induced matrix $G_{x_t} = J_{x_t}^\top J_{x_t}$ remains symmetric and positive (semi-)definite.

Geodesics under our proposed metric are obtained by minimizing Eq. (11). The objective has a simple form and is numerically stable, but it is more computationally expensive than closed-form interpolations such as LERP or SLERP. Methods based on conformal metrics (e.g., Yu et al. (2025)) also require solving an optimization problem (namely, a boundary-value problem) to interpolate between two points, and thus have comparable computational cost.

## 5 EXPERIMENTS

### 5.1 SYNTHETIC 2D DATA

To illustrate the behavior of the geodesic under our proposed metric, we first conduct experiments on a synthetic 2D dataset, shown in Fig. 1 (left) and (middle) (see Appendix C.1 for details). We constructed a C-shaped distribution on a 2D space and trained a DDPM (Ho et al., 2020) on this dataset. Then, we obtained interpolations between two points using different methods at time $t = 0.02T$ through DDIM Inversion. See Appendix B.2 for comparison methods.

LERP completely ignores the data manifold and traverses through low-density regions. SLERP follows the manifold to some extent but slightly deviates from it. Density-based interpolation is a geodesic under a conformal density-based metric proposed in Yu et al. (2025), which approaches and traverses a high-density region, not preserving the probabilities of the endpoints. A geodesic under our proposed metric runs parallel to the data manifold, preserving the probabilities of the endpoints and yielding natural transitions.

We randomly sampled 50 pairs of endpoints from the distribution and obtained interpolations using different methods. Then, we evaluated the standard deviation of the density $p(x)$ over each interpolation path and summarized the averages in Table 1. A smaller value indicates that the interpolation stays close to the data manifold, while a larger value indicates that the interpolation unnecessarily deviates from or approaches the data manifold. Geodesics under our proposed metric maintain a consistent distance from the data manifold, resulting in smoother and more coherent interpolations.

Table 1: Results on the synthetic 2D dataset.

| Methods | Std. of Prob. ↓ |
|---|---|
| LERP | 0.1606 |
| SLERP | 0.0833 |
| Density-based | 0.1073 |
| Ours | **0.0701** |

### 5.2 IMAGE INTERPOLATION

**Experimental Setup.** To evaluate our proposed Riemannian metric $g_{x_t}$, we perform image interpolation, a common proxy for assessing the quality of learned data manifolds in DGMs. This requires computing the geodesic between two images, which serves as an interpolated image sequence. We denote the original pair of images by $x_0^{(0)}$ and $x_0^{(1)}$, and the interpolated image sequence by $\{\hat{x}_0^{(u)}\}$ for $u \in [0, 1]$. We use Stable Diffusion v2.1-base (Rombach et al., 2022) as the backbone, set the number of timesteps to $T = 50$, and set the number of discretization points to $N = 10$.

We evaluate methods on three benchmark datasets: the animation subset of MorphBench (MB(A)) (Zhang et al., 2023), Animal Faces-HQ (AF) (Choi et al., 2020), and CelebA-HQ (CA) (Karras et al., 2018a). Because our goal is to capture the local geometry of the data manifold, we exclude the metamorphosis subset from MorphBench, which contains significant (i.e., global) shape changes. For each of Animal Faces-HQ and CelebA-HQ, we curate 50 image pairs with Low-Perceptual Image Patch Similarity (LPIPS) (Zhang et al., 2018) below 0.6 to ensure semantic similarity, closely following the procedure in Yu et al. (2025). Further details are provided in Appendix C.2.

**Comparison Methods.** We used the following baseline methods for comparison: LERP (Ho et al., 2020), SLERP (Song et al., 2021a), NAO (Samuel et al., 2023), NoiseDiffusion (NoiseDiff) (Zheng et al., 2024) and GeodesicDiffusion (GeoDiff) (Yu et al., 2025). We use default settings for comparison methods (NAO, NoiseDiff, GeoDiff) based on their official codes. See Appendix B.2 for more details. For LERP, SLERP, and our proposed metric, we used the DDIM Scheduler (Song et al., 2021a) and operated in the noise space at $t = 0.6T$. For our proposed metric, each path was initialized with SLERP and updated for 500 iterations using Adam optimizer (Kingma & Ba, 2015) with a learning rate of 0.001, decayed with cosine annealing to 0.0001 (Loshchilov & Hutter, 2017). We also adopted the prompt adjustment of Yu et al. (2025); see Appendix B.3.

Table 2: Image interpolation results (lower is better).

| Method | FID ↓ | | | PPL ↓ | | | PDV ↓ | | | RE ↓ (×10⁻³) | | |
|---|---|---|---|---|---|---|---|---|---|---|---|---|
| | MB(A) | CA | AF | MB(A) | CA | AF | MB(A) | CA | AF | MB(A) | CA | AF |
| LERP | 84.20 | 95.68 | 119.58 | 0.848 | 1.420 | 1.859 | 0.055 | 0.091 | 0.154 | 0.401 | 1.010 | 2.049 |
| SLERP | 62.81 | 37.84 | 26.07 | 0.644 | 0.707 | 0.871 | 0.030 | **0.033** | **0.022** | 0.401 | 1.010 | 2.049 |
| NAO | 130.54 | 83.05 | 71.47 | 2.868 | 2.121 | 2.443 | 0.163 | 0.154 | 0.173 | 39.244 | 27.623 | 40.178 |
| NoiseDiff | 119.47 | 65.04 | 68.87 | 3.618 | 2.098 | 3.250 | 0.064 | 0.069 | 0.083 | 15.096 | 8.618 | 19.628 |
| GeoDiff | 28.70 | 35.98 | 25.80 | 0.402 | 0.669 | 0.842 | 0.024 | 0.044 | 0.027 | 0.188 | 0.891 | 1.969 |
| Ours | **27.44** | **32.54** | **21.01** | **0.380** | **0.633** | **0.767** | **0.021** | 0.036 | 0.023 | **0.177** | **0.888** | **1.962** |

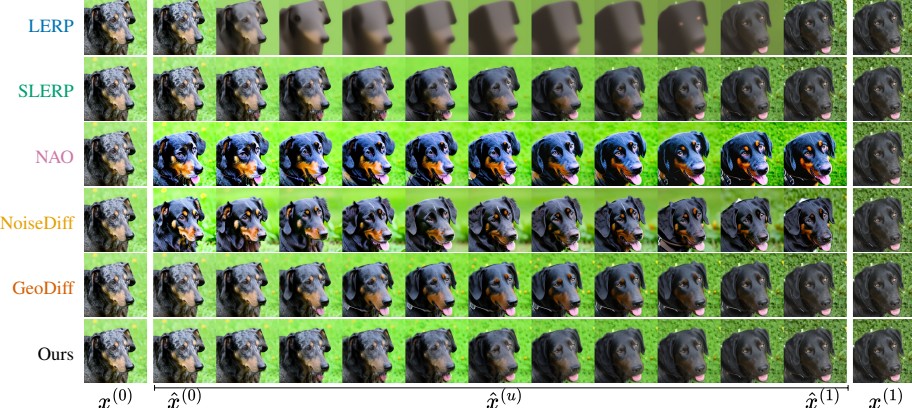

$LERP$    $SLERP$    $NAO$    $NoiseDiff$    $GeoDiff$    $Ours$

$x^{(0)}$   $\hat{x}^{(0)}$      $\hat{x}^{(u)}$      $\hat{x}^{(1)}$   $x^{(1)}$

Figure 2: Qualitative examples of interpolated image sequences for AF (Dog). The images at both ends are the given endpoints $x_0^{(0)}$ and $x_0^{(1)}$, and the middle images are the interpolated results $\{\hat{x}_0^{(u)}\}$ for $u \in [0, 1]$. See also Fig. 5 in Appendix D.

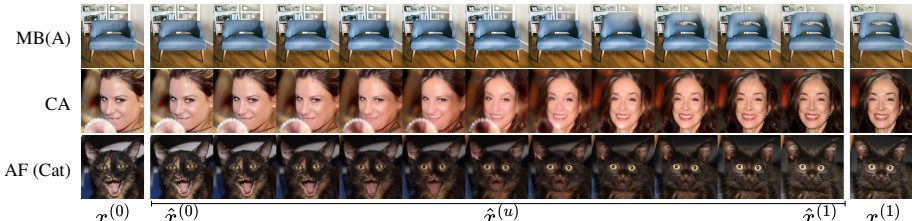

MB(A)    CA    AF (Cat)

$x^{(0)}$   $\hat{x}^{(0)}$      $\hat{x}^{(u)}$      $\hat{x}^{(1)}$   $x^{(1)}$

Figure 3: Qualitative examples of interpolated image sequences by our method.

**Metrics.** We evaluated the quality of the interpolated image sequences by the following metrics: (1) Fréchet Inception Distance (FID) (Heusel et al., 2017) measures the similarity between the set of input images and the set of interpolated images using features extracted from a pre-trained Inception v3 network (Szegedy et al., 2015); (2) Perceptual Path Length (PPL) (Karras et al., 2018b) is the sum of LPIPS between adjacent images to assess the directness of the interpolated image sequence; (3) Perceptual Distance Variance (PDV) (Zhang et al., 2023) is the standard deviation of LPIPS between adjacent images to assess the consistency of transition rates over the interpolated image sequence; and (4) Reconstruction Error (RE) is the mean squared error between the input pair of images, $x_0^{(0)}$ and $x_0^{(1)}$, and the first and last samples of the interpolated image sequence, $\hat{x}_0^{(0)}$ and $\hat{x}_0^{(1)}$, to assess how well the endpoints are preserved.

**Results.** We summarize the quantitative results in Table 2. Using geodesics under our proposed metric, image interpolation achieves the best scores on all datasets for FID, PPL, and RE. It also records the best PDV on MB(A) and the second-best on the others. Figures 2 and 3 show qualitative results. See also Fig. 5 in Appendix D for comparisons. As reported previously, LERP yields blurry interpolations. Although NAO and NoiseDiff generate high-quality images, these methods generate glossy textures that are absent in the original images and exhibit extremely large reconstruction errors, failing a proper interpolation. This is because they adjust the norms of noisy samples $x_t^{(u)}$

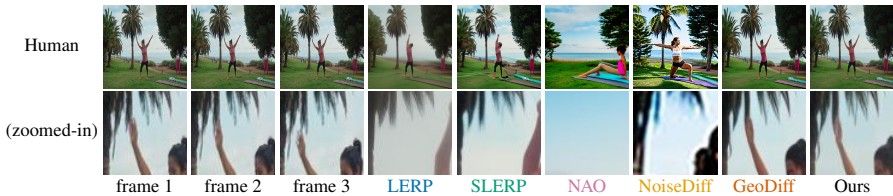

Figure 4: Qualitative examples on video frame interpolations. See also Fig. 6 in Appendix D.

in the noise space to mitigate the feature loss observed with LERP, which also alters endpoints and causes large deviations from the originals. SLERP produces sharper results than LERP but still lags behind geodesic-based methods. GeoDiff ranks second in most cases after our proposed metric, but produces unusually glossy images that lack fine details. This trend is consistent with prior reports that sample density is negatively correlated with perceptual detail (Karczewski et al., 2025a). Our metric yields interpolations that are high-quality and preserve fine details of the input images.

## 5.3 VIDEO FRAME INTERPOLATION

**Experimental setup.** Image interpolation has no ground-truth images, which limits evaluation to indirect measures. To address this, we evaluate methods on video frame interpolation using Mean Squared Error (MSE) and LPIPS against the ground-truth middle frames. We employ three benchmarks curated by Zhu et al. (2024): 21 natural-scene clips from DAVIS (Perazzi et al., 2016), 56 human-pose clips from Pexels (Human), and 26 indoor/outdoor clips selected from RealEstate10K (RE10K) (Zhou et al., 2018). We use three consecutive frames from each video clip: we take frames 1 and 3 as $x_0^{(0)}$ and $x_0^{(1)}$ and estimate frame 2 as $\hat{x}_0^{(0.5)}$. This is because the interpolation between two frames that are far apart in time may not be unique, which is not suitable for comparison with ground-truth frames. Unless otherwise specified, all methods and hyperparameters are identical to those used for image interpolation. Each frame is resized to $512 \times 512$ pixels. We generate a text prompt from frame 1 for each clip using BLIP-2 (Li et al., 2022).

**Results.** Table 3 summarizes the quantitative results. Our method achieves the lowest MSE and LPIPS on all datasets. Figures 4 and 6 provide qualitative results. As shown in zoomed-in images, only ours and GeoDiff interpolate the arm movement well. LERP produces blurry outputs, which is consistent with its poor LPIPS score despite a relatively low MSE. NAO and NoiseDiff produce images with large deviations from the ground-truth frames. SLERP sometimes fails to preserve small objects and textures: a person's arm and background objects on Human, water ripples and a bird's neck on DAVIS, and small furniture on RE10K.

Table 3: Video frame interpolation results.

| Method | MSE ↓ ($\times 10^{-3}$) | | | LPIPS ↓ | | |
|---|---|---|---|---|---|---|
| | DAVIS | Human | RE10K | DAVIS | Human | RE10K |
| LERP | 12.135 | 4.566 | 6.299 | 0.590 | 0.379 | 0.377 |
| SLERP | 15.440 | 6.080 | 6.128 | 0.487 | 0.320 | 0.301 |
| NAO | 108.211 | 99.867 | 121.680 | 0.679 | 0.668 | 0.664 |
| NoiseDiff | 46.881 | 41.994 | 28.867 | 0.561 | 0.552 | 0.482 |
| GeoDiff | 13.253 | 3.363 | 5.941 | 0.334 | 0.184 | 0.229 |
| Ours | **8.777** | **2.018** | **2.771** | **0.318** | **0.170** | **0.178** |

GeoDiff yields relatively coherent results, but it increases color saturation and over-smooths textures (e.g., flattened water ripples), which indicates a loss of fine details. Overall, our method preserves edges, object shapes, and textures more faithfully than the others.

See Appendix D for more qualitative results, ablation study, and visualizations.

## 6 CONCLUSION

In this paper, we introduced a novel Riemannian metric, inspired by recently found link between the Jacobian of the score function and the local structure of the data manifold learned by diffusion models. Our proposed metric encourages geodesics to stay within or run parallel to the data manifold, yielding natural transitions that preserve the underlying semantics, as verified through experiments on synthetic 2D data, image interpolation, and video frame interpolation. Applications to other metric-related tasks, such as clustering, are left for future work.

## ETHICS STATEMENT

This study is purely focused on analysis of diffusion models, and it is not expected to have any direct negative impact on society or individuals.

## REPRODUCIBILITY STATEMENT

The environment, datasets, methods, evaluation metrics, and other experimental settings are given in Section 5 and Appendices B and C. For full reproducibility, the source code is attached as supplementary material.

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

# A   DETAILED EXPLANATIONS

## A.1   LOCAL COORDINATE EXPRESSION

A Riemannian metric $g$ is symmetric and positive-definite; i.e., at $p \in \mathcal{M}$,

$$g_p(v, w) = g_p(w, v), \quad g_p(v, v) \geq 0 \text{ for all } v \in \mathcal{T}_p\mathcal{M}, \quad g_p(v, v) = 0 \Leftrightarrow v = 0.$$

By identifying $g_p$ with an inner product, we write

$$\langle v, w \rangle_g := g_p(v, w) \quad \text{for any } v, w \in T_p\mathcal{M}.$$

Let $(x^1, \ldots, x^D)$ be smooth local coordinates in a neighborhood of $p \in \mathcal{M}$. Then, the coordinate basis for $T_p\mathcal{M}$ is $\left(\frac{\partial}{\partial x^1}|_p, \ldots, \frac{\partial}{\partial x^D}|_p\right)$, where $\frac{\partial}{\partial x^i}$ is the $i$-th coordinate vector field. Tangent vectors $v, w \in T_p\mathcal{M}$ can be expressed as $v = \sum_{i=1}^{D} v^i \frac{\partial}{\partial x^i}|_p$ and $w = \sum_{i=1}^{D} w^i \frac{\partial}{\partial x^i}|_p$, respectively. The matrix notation $G_p$ of $g$ at $p$ consists of $(i, j)$-elements

$$g_{ij}(p) = g_p\left(\frac{\partial}{\partial x^i}|_p, \frac{\partial}{\partial x^j}|_p\right) = \left\langle \frac{\partial}{\partial x^i}|_p, \frac{\partial}{\partial x^j}|_p \right\rangle_g \tag{12}$$

for $i, j = 1, 2, ..., D$. The Euclidean metric is represented by the identity matrix $I$. The inner product of $v$ and $w$ with respect to the Riemannian metric $g_p$ is

$$g_p(v, w) = \sum_{i=1}^{D} \sum_{j=1}^{D} g_{ij}(p) v^i w^j = v^T G_p w. \tag{13}$$

## A.2   EXPLANATION OF PROPOSITION 1

When the score function $s_\theta$ is exact, it is the gradient $\nabla_{x_t} \log p_t(x_t; \theta)$ of the log-density $\log p_t(x_t; \theta)$, and its Jacobian $J_{x_t}$ equals the Hessian, $J_{x_t} = \nabla_{x_t} \nabla_{x_t} \log p_t(x_t; \theta)$, which is symmetric. In this idealized case, its eigenvectors form an orthonormal basis of the noise space $\mathbb{R}^D$. We divide these eigenvectors into a basis for the tangent space $\mathcal{T}_x\mathcal{M}_t$, $\{v_i\}_{i=1}^{d}$ (with small eigenvalues $\lambda_i$), and a basis for the normal space $\mathcal{N}_x\mathcal{M}_t$, $\{v_j\}_{j=d+1}^{D}$ (with large eigenvalues $\lambda_j$). These spaces are orthogonal complements of each other, and the tangent space $\mathcal{T}_x\mathbb{R}^D$ to the noise space $\mathbb{R}^D$ at $x$ can be decomposed into their direct sum, $\mathcal{T}_x\mathbb{R}^D = \mathcal{T}_x\mathcal{M}_t \oplus \mathcal{N}_x\mathcal{M}_t$. Any tangent vector $v \in \mathcal{T}_x\mathbb{R}^D$ is uniquely decomposed as $v = v_\mathcal{T} + v_\mathcal{N}$, where $v_\mathcal{T} \in \mathcal{T}_x\mathcal{M}_t$ and $v_\mathcal{N} \in \mathcal{N}_x\mathcal{M}_t$. The squared Jacobian-vector product $\|J_{x_t} v\|_2^2$ can be expanded as:

$$\|J_{x_t} v\|_2^2 = \|J_{x_t}(v_\mathcal{T} + v_\mathcal{N})\|_2^2 = \|J_{x_t} v_\mathcal{T}\|_2^2 + \|J_{x_t} v_\mathcal{N}\|_2^2 + 2\langle J_{x_t} v_\mathcal{T}, J_{x_t} v_\mathcal{N} \rangle. \tag{14}$$

Due to the orthogonality of the eigenspaces, the cross term $\langle J_{x_t} v_\mathcal{T}, J_{x_t} v_\mathcal{N} \rangle$ vanishes, and we have

$$\begin{aligned} \|J_{x_t} v_\mathcal{T}\|_2^2 &= \sum_{i=1}^{d} \lambda_i^2 \langle v, v_i \rangle^2 \approx 0, \\ \|J_{x_t} v_\mathcal{N}\|_2^2 &= \sum_{j=d+1}^{D} \lambda_j^2 \langle v, v_j \rangle^2 \gg 0 \quad (\text{if } v_\mathcal{N} \neq 0). \end{aligned} \tag{15}$$

Hence, minimizing the squared Jacobian–vector product $\|J_{x_t} v\|_2^2$ (under a fixed Euclidean norm of $v$) is dominated by minimizing the normal-space component $\|J_{x_t} v_\mathcal{N}\|_2^2$, and essentially encourages the vector $v$ to lie in the tangent space $\mathcal{T}_x\mathcal{M}_t$.

In practice, diffusion models learn the score function $s_\theta$ directly, its Jacobian $J_{x_t}$ need not be symmetric, and the right singular vectors need not be exactly orthogonal to each other. Even then, minimizing $\|J_x v\|_2^2$ still suppresses the component in the subspace spanned by the large right singular vectors and amplifies the component spanned by the small right singular vectors; Proposition 1 continues to hold in this generalized sense.

# B   EXPERIMENTAL SETUP

## B.1   DDIM INVERSION

Naive encoding of an original image is to add Gaussian noise as in the forward process $q(x_t \mid x_{t-1})$, which is stochastic and often yields poor reconstructions. To accurately invert the reverse process and recover the specific noise map associated with a given image, *DDIM Inversion* (Mokady et al.,

2023) is widely used. The deterministic version ($\eta = 0$) of DDIM can be regarded as an ordinary differential equation (ODE) solved by the Euler method (Song et al., 2021a;b). In the limit of infinitesimally small timesteps, the ODE is invertible.

Concretely, setting $\sigma_t = 0$ in Eq. (6) gives

$$x_{t-1} = a_t x_t + b_t \epsilon_\theta(x_t, t) = x_t + (a_t - 1)x_t + b_t \epsilon_\theta(x_t, t), \tag{16}$$

where $a_t = \sqrt{\alpha_{t-1}/\alpha_t}$ and $b_t = -\sqrt{\alpha_{t-1}(1 - \alpha_t)/\alpha_t} + \sqrt{1 - \alpha_{t-1}}$. This can be viewed as an ODE with the time derivative $(a_t - 1)x_t + b_t \epsilon_\theta(x_t, t)$ solved by the Euler method with the unit step size. With a sufficiently small timestep size,

$$x_t = \frac{x_{t-1} - b_t \epsilon_\theta(x_t, t)}{a_t} \approx \frac{x_{t-1} - b_t \epsilon_\theta(x_{t-1}, t)}{a_t}, \tag{17}$$

since $\epsilon_\theta(x_t, t) \approx \epsilon_\theta(x_{t-1}, t)$. The deterministic forward process iteratively applies the update rule in Eq. (17) to a sample $x_0$ from $t = 0$ to $\tau$ and obtains the noisy image $x_\tau$, from which the deterministic reverse process reconstructs the original $x_0$ up to numerical errors. This inversion procedure substantially improves the fidelity of reconstructions and subsequent interpolations.

## B.2 COMPARISON METHODS

**Linear Interpolation.** Once samples are noised via DDIM Inversion, one can perform straightforward linear interpolation (LERP) (Ho et al., 2020), by treating the noise space at fixed time $t = \tau > 0$ as a linear latent space. Given samples $x_0^{(0)}$ and $x_0^{(1)}$ in the data space, the deterministic forward process obtains their noised versions $x_\tau^{(0)}$ and $x_\tau^{(1)}$ at $\tau$, respectively. A linear interpolation in that space is given by

$$x_\tau^{(u)} = (1 - u)x_\tau^{(0)} + u x_\tau^{(1)}, \tag{18}$$

where $u \in [0, 1]$ is the interpolation parameter. Then, one applies the deterministic reverse process from $t = \tau$ back to $t = 0$ to obtain a sequence of interpolated images $x_0^{(u)}$ in the data space.

**Spherical Linear Interpolation.** An alternative is spherical linear interpolation (SLERP) (Song et al., 2021a), which finds the shortest path on a sphere in the noise space:

$$x_\tau^{(u)} = \frac{\sin((1-u)\theta)}{\sin(\theta)} x_\tau^{(0)} + \frac{\sin(u\theta)}{\sin(\theta)} x_\tau^{(1)} \tag{19}$$

where $\theta = \arccos\left(\frac{(x_\tau^{(0)})^\top x_\tau^{(1)}}{\|x_\tau^{(0)}\| \|x_\tau^{(1)}\|}\right)$. This procedure preserves the norms of the noisy samples $x_\tau^{(u)}$, yielding more natural interpolations than LERP. Note that SLERP assumes that $x_\tau^{(0)}$ and $x_\tau^{(1)}$ are drawn from a normal distribution, which holds only for a sufficiently large $t$ (typically, $t = T$). Nonetheless, SLERP is often applied at moderate $t$.

## B.3 PROMPT ADJUSTMENT

To improve the quality of interpolations, we adopt the prompt adjustment proposed by Yu et al. (2025). Internally in Stable Diffusion v2.1-base (Rombach et al., 2022), a text prompt $c$ is first encoded into a text embedding $z$ using CLIP (Radford et al., 2021). To better align the text embedding $z$ with a given pair of images $x_0^{(0)}$ and $x_0^{(1)}$, we adjust the text embedding $z$ in a similar way to textual inversion (Gal et al., 2023). Namely, the text embedding $z$ is updated to minimize the DDPM loss in Eq. (5) for 500 iterations for image interpolation and 1,000 iterations for video frame interpolation. We use AdamW optimizer (Loshchilov & Hutter, 2019) with a learning rate of 0.005.

Also following Yu et al. (2025), we do not use CFG (i.e., set $w = 0$ in Eq. (8)) but use the following negative prompt $c_{\text{neg}}$ with $w_{\text{neg}} = 1$: "A doubling image, unrealistic, artifacts, distortions, unnatural blending, ghosting effects, overlapping edges, harsh transitions, motion blur, poor resolution, low detail."

## C DETAILS OF EXPERIMENTS

This section provides additional details of the experiments in Section 5. All experiments were conducted on a single NVIDIA A100 GPU.

### C.1 DETAILS OF SYNTHETIC 2D DATASET

**Dataset.**    We construct a two-dimensional C-shaped distribution as follows. We start with an axis-aligned ellipse with semi-axes 1.0 (along $x_1$) and 1.2 (along $x_2$). To open the "C", we remove all points in a $\pm 30°$ wedge centered on the positive $x_1$-axis. We then add isotropic Gaussian perturbations with standard deviation 0.001 per coordinate to each point. From the resulting distribution, we draw 100,000 samples.

**Network.**    The noise predictor $\epsilon_\theta$ is composed of three linear layers of hidden width 512 with SiLU activation functions (Elfwing et al., 2017). The network takes a tuple of a data point $x$ and a normalized time $t$ as input. We set the number of steps to $T = 1,000$. We trained this network for 1,000 epochs using the AdamW optimizer (Loshchilov & Hutter, 2019) with a batch size of 512. The learning rate follows cosine annealing (Loshchilov & Hutter, 2017), decaying from $10^{-3}$ to 0 without restarts. For stability, we apply gradient-norm clipping with a threshold of 1.0.

**Implementation Details.**    In Fig. 1 (left), we visualize the interpolation between $x_0^{(0)} = (0.0, 1.15)$ and $x_0^{(1)} = (-0.8, -0.6)$ with $N = 100$ discretization points. Comparison methods include Linear Interpolation (LERP) (Ho et al., 2020), Spherical Linear Interpolation (SLERP) (Song et al., 2021a), and density-based interpolation based on the metric proposed in Yu et al. (2025). We used the DDIM Scheduler (Song et al., 2021a) and operated in the noise space at $t = 0.02T$. For our method and the density-based interpolation, we find the geodesic paths by minimizing the energy functional $E[\gamma]$. Both paths are initialized using SLERP and updated using Adam optimizer (Kingma & Ba, 2015) for 1,000 iterations with a learning rate of $10^{-4}$.

### C.2 DATASETS FOR IMAGE INTERPOLATION

The animation subset of MorphBench (Zhang et al., 2023) is a dataset of pairs of images obtained via image editing. Each pair is associated with a text prompt; we used the provided prompts as the condition $c$.

Animal Faces-HQ (Choi et al., 2020) is a dataset of high-resolution images of animal faces. From this dataset, we randomly selected 50 pairs of dog images and 50 pairs of cat images with LPIPS below 0.6 to ensure semantic similarity. We used the text prompts "a photo of a dog" for dog images and "a photo of a cat" for cat images.

CelebA-HQ (Karras et al., 2018a) is a high-resolution dataset of celebrity faces. We randomly sampled 50 male pairs and 50 female pairs, again with LPIPS less than 0.6, and condition on "a photo of a man" and "a photo of a woman," respectively.

## D ADDITIONAL RESULTS

### D.1 ADDITIONAL QUALITATIVE RESULTS FOR IMAGE AND VIDEO FRAME INTERPOLATION

In this section, we provide additional qualitative results. Figures 5 and 6 provide more examples of image interpolation and video frame interpolation, which complement Fig. 2 and Fig. 4 in the main text, respectively.

### D.2 ABLATION STUDY

We adopt the prompt adjustment of GeoDiff (Yu et al., 2025) to better align the text embedding with the images. Table 4 reports an ablation on video frame interpolation. Because GeoDiff is designed to operate with this adjustment enabled, we do not report a GeoDiff variant without it. With the adjustment, both our metric and SLERP improve in MSE and LPIPS. The gains are larger for our metric: the adjustment enables the guided diffusion model to better capture the local data manifold, and our metric explicitly leverages such refined local information. By contrast, SLERP focuses on the Gaussian prior and is less sensitive to refinements.

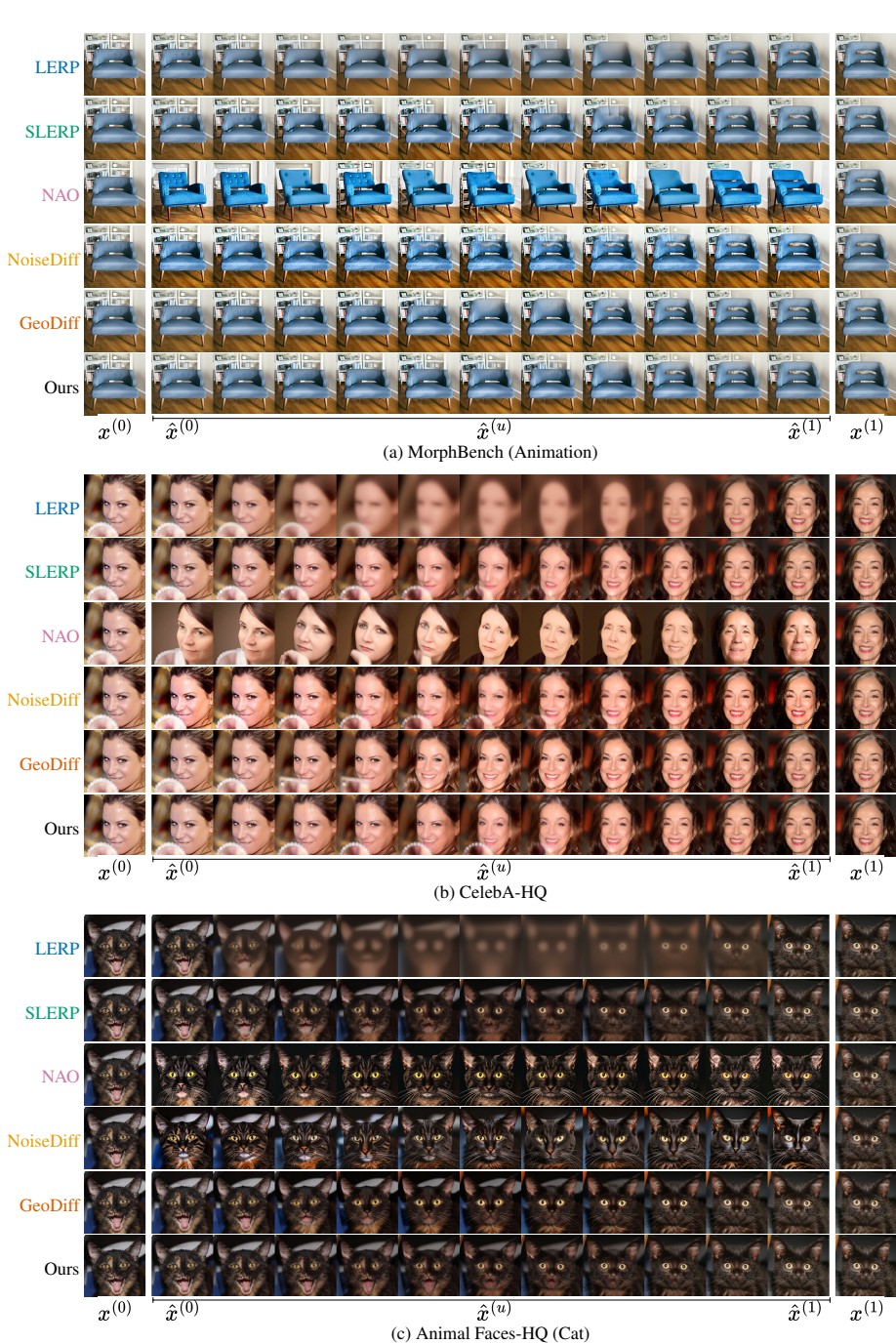

Figure 5: Examples of interpolated image sequences. The leftmost and rightmost images are the given endpoints $x_0^{(0)}$ and $x_0^{(1)}$, and the middle images are the interpolated results $\{\hat{x}_0^{(u)}\}$ for $u \in [0, 1]$.

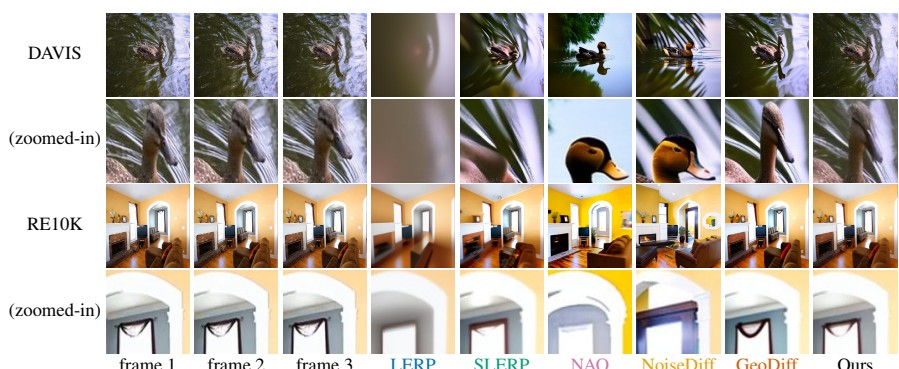

DAVIS

(zoomed-in)

RE10K

(zoomed-in)

frame 1    frame 2    frame 3    LERP    SLERP    NAO    NoiseDiff    GeoDiff    Ours

Figure 6: Qualitative examples on video frame interpolations

Table 4: Ablation study on prompt adjustment.

| Method | Adj. | MSE ↓ $(\times 10^{-3})$ | | | LPIPS ↓ | | |
| | | DAVIS | Human | RE10K | DAVIS | Human | RE10K |
|---|---|---|---|---|---|---|---|
| SLERP | | 15.440 | 6.080 | 6.128 | 0.487 | 0.320 | 0.301 |
| SLERP | ✓ | 9.894 | 2.559 | 3.778 | 0.355 | 0.200 | 0.200 |
| GeoDiff | ✓ | 13.253 | 3.363 | 5.941 | 0.334 | 0.184 | 0.229 |
| Ours | | 13.517 | 5.008 | 6.016 | 0.500 | 0.350 | 0.325 |
| Ours | ✓ | 8.777 | 2.018 | 2.771 | 0.318 | 0.170 | 0.178 |

## D.3 INTERPOLATIONS AND SPECTRAL GAPS WITH VARYING $\tau$

We visualize interpolations for varying time $\tau$ for the noise space in Fig. 7. At $\tau = 0$, intermediate samples exhibit artifacts. With no injected noise, the data manifold is extremely thin, and finding a geodesic under our metric becomes ill-conditioned. As $\tau$ increases, the interpolations become smoother and more globally coherent. At $\tau = T$, however, the interpolations are no longer semantically coherent: the noisy-sample distribution is close to Gaussian, the data manifold is not well defined, and meaningful geodesics cannot be recovered. Empirically, $\tau \in [0.4T, 0.6T]$ yields the best visual quality.

Figure 8 shows the singular values of the Jacobian $J_{x_t}$ of the score function $s_\theta$ at the point $x_\tau^{(1)}$ obtained by DDIM inversion to the rightmost image $x_0^{(1)}$. Stable Diffusion v2.1-base (Rombach et al., 2022) operates VAE's latent space of $64 \times 64 \times 4 = 16,384$ dimensions. Across timesteps, hundreds of singular values are near zero, suggesting a local intrinsic dimensionality on the order of a few hundred. As $\tau$ increases, more singular values approach 1.0 because the injected noise thickens the manifold and makes it isotropic.

## USE OF LARGE LANGUAGE MODELS.

We used ChatGPT and GitHub Copilot as autocomplete tools in polishing the manuscript and implementing the experimental code. No large language models were used for research ideation.

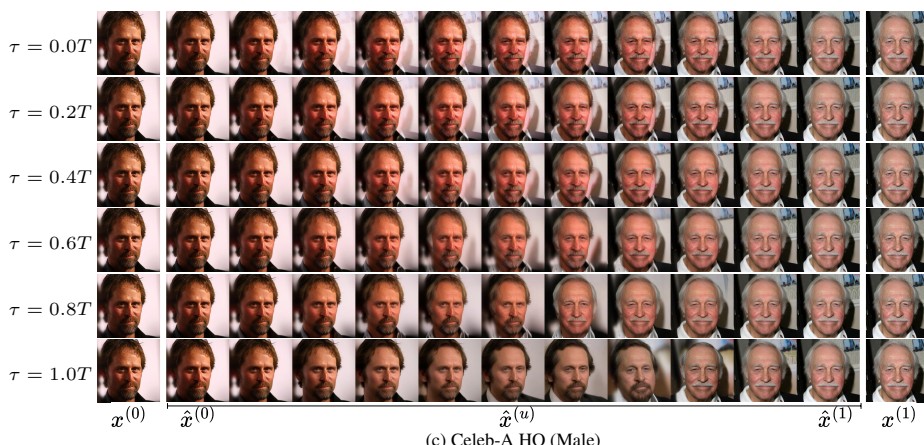

$x^{(0)}$  $\hat{x}^{(0)}$  $\hat{x}^{(u)}$  $\hat{x}^{(1)}$  $x^{(1)}$

(c) Celeb-A HQ (Male)

Figure 7: Qualitative examples of interpolated image sequences with different $\tau$. The leftmost and rightmost images are the given endpoints $x_0^{(0)}$ and $x_0^{(1)}$, and the middle images are the interpolated results $\{\hat{x}_0^{(u)}\}$ for $u \in [0, 1]$.

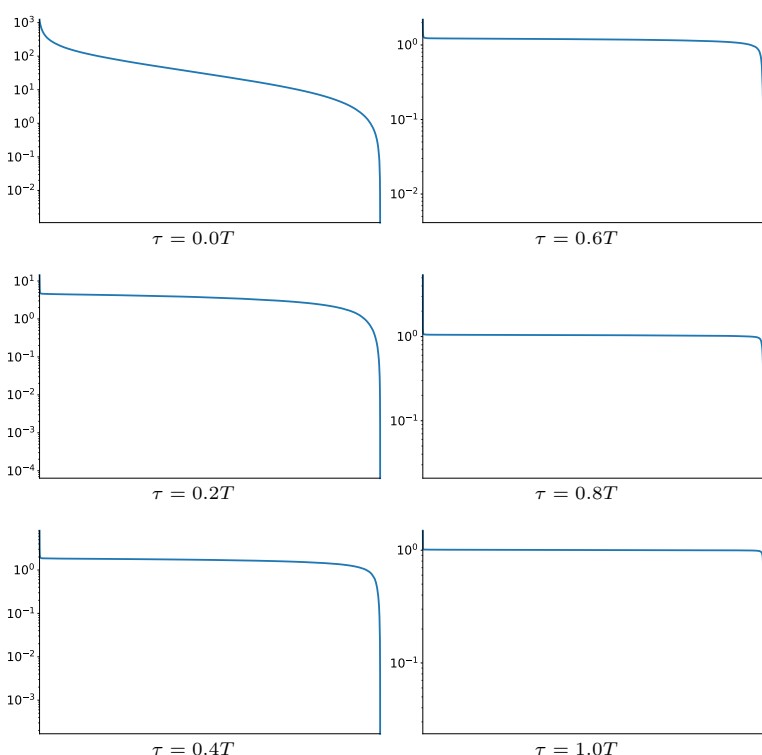

Figure 8: Examples of singular value spectra of the Jacobian $J_{x_t}$ of the score function $s_\theta$ at the right endpoint $x_0^{(1)}$ of the interpolation shown in Fig. 7, with different $\tau$. The horizontal and vertical axes represent the index and the singular value (in log scale), respectively.

