# OpenReview forum: "Be Tangential to Manifold: Discovering Riemannian Metric for Diffusion Models"
_ICLR.cc/2026/Conference — ICLR 2026 Conference Withdrawn Submission_

### Official Review · Reviewer_jhnn · 2025-10-26

**Soundness:** 3
**Presentation:** 3
**Contribution:** 1
**Rating:** 2
**Confidence:** 5

**Summary:**

This paper proposes a Riemannian metric for diffusion model interpolation using the Jacobian of the score function. Since diffusion models lack explicit latent spaces, current interpolation methods either cut through low-density regions (linear) or produce over-smoothed results (density-based). The metric leverages the finding that the score function's Jacobian captures local manifold structure - small along tangent directions, large along normal directions. Geodesics under this metric run parallel to the data manifold rather than through high-density regions.

**Strengths:**

1. The paper demonstrates expertise and deep engagement with the topic by building on findings from multiple papers.

2. Good writing quality.

3. The video-related metrics are quite impressive.

**Weaknesses:**

1. Lack of novelty - The proposed Riemannian metric does not appear particularly novel. Similar concepts are already proposed by other papers cited below. The interpolation method of minimizing the energy function based on this metric appears too simple to claim novelty.

2. It would be essential to include more baselines like the Isometric Diffusion model [1] and DiffMorpher [2].

[1] Isometric Representation Learning for Disentangled Latent Space of Diffusion Models (ICML 2024)

[2] DiffMorpher: Unleashing the Capability of Diffusion Models for Image Morphing (CVPR 2024)

**Questions:**

1. In Figure 1 middle panel, is $u$ the interpolation coefficient?

2. In Figure 1 middle panel, it is unclear why maintaining constant $p(x)$ is the geodesic. (I roughly understand but am unsure if it is always true.) Even if the rightmost images are cherry-picked, it is hard to see that "ours" is more natural or superior to SLERP or density-based methods.

3. Method line 239 defines $v$ as being in the tangent space of $x_t$. On the other hand, line 258 says "if $v$ lies in the tangent space to the data manifold." Since $x_t$ is already on the data manifold $M_t$, shouldn't $v$ always be in the tangent space according to the earlier definition?

4. At line 256, it says $s_θ(x_t,t)$ is orthogonal to the data manifold as $t\rightarrow0$. Doesn't this mean Proposition 1 only holds when $t$ is small?

5. I understand this is for interpolation on already trained models - is this the same for other methods? If so, how do you measure FID for each method? If FID only measures the endpoints, what does it mean? I am curious about the same thing on Reconstruction Error.

6. In my knowledge, PPL and PDV metrics are quite noisy, so they need to be measured with large sample sizes or should be reported with standard deviations. How many samples were used to measure those? Also, since "ours" performs worse than SLERP on PDV for some datasets, couldn't we argue that SLERP produces more naturally smooth transitions, as suggested by the lower variance?

---

### Official Review · Reviewer_3QED · 2025-10-29

**Soundness:** 3
**Presentation:** 3
**Contribution:** 4
**Rating:** 6
**Confidence:** 3

**Summary:**

The author assume generative model must be modeling the low dimensional manifold of structured high dimensional data. However, the modeling of low-dimensional data is implicit so it doesn't give us a way to explore the geometric structure of the manifold, travel geodesically between two point on the manifold. The author use the framework of Riemannian metric to study the manifold structure modeled by diffusion model, and provided algorithm to travel on the manifold, leading to real world application like image interpolation.

**Strengths:**

Good paper. This is a very a nature idea. I am happy there's finally an extensive study on this.

**Weaknesses:**

I think this work is related and should be cited [1] They did manifold extrapolation instead of interpolation. The evaluation metric used in this paper for measuring extrapolation quality could be relevant. You want to calculate the distance between interpolated video and ground truth video, the frame might not match frame to frame, so something like dynamic time warping might be useful for evaluation.

[1] Denoising for Manifold Extrapolation, Yun et al 2024

**Questions:**

You cite [1]. I remember they found something like the manifold structure in the latent space of VAE is surprisedly flat. In some sense, the VAE is flattening the manifold of natural images. Just curious, do you think you can do similar analyze and find similar result for diffusion model?


The riemannian geometry of deep generative models, Hang Shao et al, 2018

---

### Official Review · Reviewer_Ng2X · 2025-10-31

**Soundness:** 3
**Presentation:** 3
**Contribution:** 2
**Rating:** 4
**Confidence:** 4

**Summary:**

This paper proposes a novel Riemannian metric for diffusion models that leverages the Jacobian of the score function to define a local inner product on the noise space. The goal is to construct geodesics that remain within or parallel to the data manifold, enabling semantically faithful interpolations in diffusion-based generative models. The authors argue that the Jacobian encodes tangent and normal directions to the learned data manifold, and thus, a Riemannian metric built from it (Eq. 9: $g_{x_{t}}(v,w) = v^{T}J_{x_{t}}^{T} J_{x_{t}} w$) captures the intrinsic geometry of the data.
Empirical results demonstrate improvements over linear, spherical, and density-based interpolation methods in terms of FID, PPL, and reconstruction error on several benchmarks (Animal Faces-HQ, CelebA-HQ, MorphBench) and video frame interpolation tasks. The method does not require retraining or architectural changes to the diffusion model.

**Strengths:**

- The paper is clearly written and easy to follow.
- The paper includes quantitative and qualitative experiments on both synthetic 2D data and realistic image/video benchmarks. The video interpolation results are particularly interesting.

**Weaknesses:**

- The motivation for the proposed metric $g\_{x\_{t}} (v,w) = v^{T} J\_{x\_{t}}^{T} J\_{x\_{t}} w$ remains insufficient. The discussion in Lines 245–248 mainly explains the Gram-matrix structure $J\_{x\_{t}}^{T} J\_{x\_{t}}$, but not why the Jacobian of the score function is the most appropriate choice. A plausible alternative would be the score-based metric $g_{x_{t}}(v,w) = v^{T} s_{\theta}(x_{t}, t) s_{\theta}(x_{t}, t)^{T} w$, which is also positive semi-definite. The paper should clarify why the Jacobian provides a superior geometric characterization.
- The justification for introducing the Jacobian of the score function in defining the Riemannian metric relies heavily on the Jacobian’s spectral property observed in Ventura et al. (2025) without sufficient in-paper evidence. Since this is the main justification of the proposed metric design, the supporting experiment visualizing or quantifying the singular value distribution should be included in the main text rather than included in Appendix Fig. 8.
- In Lines 256-258, it is commented that Stanczuk et al. (2024) demonstrated that the score function itself exhibits similar tangent/normal decomposition behavior: the inner product between the score function $s_{\theta}(x_{t}, t)$ and a vector $v$ at $x_{t}$ is small if $v$ lies in the tangent space to the data manifold, while it is large if $v$ points in the normal direction. In that case, the benefit of using the Jacobian-based metric over its score-based variant remains unclear. The manuscript should provide either a theoretical explanation or an empirical comparison with this baseline [1].
- Figure 8 shows that the Jacobian’s singular-value distribution depends on the noise level $t$. Quantitative analyses of how these spectral characteristics vary with $t$ and how they relate to the interpolation performance in Tables 2 and 3 would strengthen the argument.
- The geodesic optimization in Eq. (11) involves iterative updates and is computationally heavier than closed-form interpolations. The paper should quantify the runtime overhead.

[1] Azeglio, Simone, and Arianna Di Bernardo. "What's Inside Your Diffusion Model? A Score-Based Riemannian Metric to Explore the Data Manifold." Arxiv.

**Questions:**

- In Lines 275-284, what is the precise meaning of interpreting the proposed metric as the pullback of the Euclidean metric on the score space? Earlier studies have shown that gradients of log-likelihoods with respect to model parameters. However, the proposed formulation involves gradients with respect to $x_{t}$​. How should this difference be understood?
- Beyond interpolation and visualization, what practical applications can benefit from the discovered smooth geodesic structure?

---

### Official Review · Reviewer_ayqt · 2025-11-03

**Soundness:** 3
**Presentation:** 3
**Contribution:** 2
**Rating:** 4
**Confidence:** 5

**Summary:**

This paper proposes a manifold-aware interpolation method for diffusion models by defining a Riemannian metric on the noise space using the Jacobian of the score function. By minimizing the geodesic energy under this metric, the resulting interpolation paths tend to follow the underlying data manifold, producing more natural and detail-preserving transitions compared to prior approaches.

**Strengths:**

1. Well-designed evaluation protocol
The paper addresses a long-standing challenge in diffusion-based interpolation: evaluating whether the interpolated trajectory remains semantically meaningful. I appreciate that the authors go beyond commonly reported metrics such as FID and RE, and additionally incorporate PPL and PDV to assess the smoothness and consistency of the interpolation trajectory. Furthermore, the inclusion of video frame interpolation (evaluated using MSE and LPIPS) provides a grounded and objective benchmark with clear reference frames. The experimental setup is described clearly enough that the evaluation pipeline could be readily adopted in future work, which I consider a strong point of the paper.

2. Strong empirical performance
Both the quantitative results and qualitative visualizations demonstrate clear improvements over prior baseline interpolation methods. The proposed geodesic-based interpolation produces transitions that are perceptually more natural and detail-preserving, which strengthens the empirical claim that the proposed metric meaningfully reflects the underlying data geometry.

**Weaknesses:**

1. Concerns regarding the use of the VAE latent space
All main experiments are conducted using Stable Diffusion v2.1, where diffusion operates in the compressed latent space of a VAE. This means the data manifold here is already partially shaped by the VAE encoder prior to diffusion. As a result, it is unclear whether the proposed manifold-based interpolation method fundamentally applies to diffusion models in a general sense, especially those that operate directly in pixel space or other VAEs (VA-VAE,...) or in other modalities (e.g., audio, 3D, text embeddings). The fact that interpolation at \tau = 0.0T still works to some extent (Appendix Fig. 7) may stem from the geometric structure induced by the VAE, rather than purely from the diffusion process itself. Additional experiments on pixel-space diffusion models or non-image domains would strengthen the generality claim.

2. Placement and depth of Section D.3 (Interpolations and Spectral Gaps with varying \tau)
The core narrative of the paper suggests that geodesic interpolation is meaningful across timesteps. However, the empirical discussion of how the spectral structure of the Jacobian changes with \tau appears only briefly in Appendix D.3, despite being central to the theoretical motivation. For example, the statement “As \tau increases, more singular values approach 1.0 because injected noise thickens the manifold and makes it isotropic.” raises a substantial question: if the manifold becomes isotropic at larger noise levels, do geodesic paths cease to carry meaningful geometric information at those stages? This deserves more careful treatment, ideally with quantitative curves across varying \tau, possibly supported by toy experiments that illustrate when and why the proposed metric is effective.

3. Limited diversity in data domains and styles
The experiments primarily focus on natural images of faces and animals. It remains unclear how the method behaves under more complex compositions (e.g., multi-object scenes), stylized domains (e.g., cartoons, anime, drawings), or domains with high structural variation. Even brief qualitative or failure-case discussions could help clarify whether the proposed metric generalizes or whether failure modes arise from, for instance, overlapping local tangent spaces or insufficient Jacobian conditioning. Understanding such limitations would help clarify the scope of applicability.

4. (Minor) Computational Cost Clarification
It would be helpful if the paper provided computational cost.

**Questions:**

The points below reiterate the weaknesses discussed above, but I list them here for clarity and direct response.
1. Generality beyond the VAE latent space:
Can the authors clarify whether the proposed method is expected to generalize to pixel-space diffusion models or other modalities, given that all experiments are conducted in a compressed VAE latent space?

2. Effect of noise level \tau:
The appendix suggests that increasing \tau makes the manifold more isotropic. Does this reduce the meaningfulness of geodesic interpolation at higher noise levels? If possible, please provide clarification or additional quantitative trends across different \tau values.

3. Data diversity and failure modes:
Have the authors tested the method on more complex scenes or stylized domains (e.g., multi-object, cartoon/anime)? If so, what behaviors or limitations were observed?

4. Computational overhead:
Could the authors report approximate computation time and iteration counts for the geodesic optimization to better assess practical feasibility?

---

### Note · Authors · 2025-11-14

**Comment:**

We sincerely thank the reviewers for their time and effort in providing valuable feedback on our manuscript. After careful consideration of their comments, we have decided to withdraw our paper.

**Withdrawal Confirmation:**

I have read and agree with the venue's withdrawal policy on behalf of myself and my co-authors.